# Rnd3 Is a Crucial Mediator of the Invasive Phenotype of Glioblastoma Cells Downstream of Receptor Tyrosine Kinase Signalling

**DOI:** 10.3390/cells11233716

**Published:** 2022-11-22

**Authors:** Beatriz Almarán, Guillem Ramis, Silvia Fernández de Mattos, Priam Villalonga

**Affiliations:** 1Cancer Cell Biology Laboratory, Institut Universitari d’Investigació en Ciències de la Salut (IUNICS), Institut d’Investigació Sanitària Illes Balears (IdISBa), Universitat de les Illes Balears, 07122 Palma, Illes Balears, Spain; 2Serveis Científico-Tècnics, Universitat de les Illes Balears, 07122 Palma, Illes Balears, Spain; 3Departament de Biologia Fonamental i Ciències de la Salut, Universitat de les Illes Balears, 07122 Palma, Illes Balears, Spain

**Keywords:** glioblastoma, invasion, Rho GTPases, Rnd3, receptor tyrosine kinases, EGFR, actin cytoskeleton, cell motility

## Abstract

Enhanced invasiveness is one of the defining biological traits of glioblastoma cells, which exhibit an infiltrative nature that severely hinders surgical resection. Among the molecular lesions responsible for GBM aggressiveness, aberrant receptor tyrosine kinase (RTK) signalling is well-characterised. Enhanced RTK signalling directly impacts a myriad of cellular pathways and downstream effectors, which include the Rho GTPase family, key regulators of actin cytoskeletal dynamics. Here, we have analysed the functional crosstalk between oncogenic signals emanating from RTKs and Rho GTPases and focused on the specific contribution of Rnd3 to the invasive phenotype of GBM in this context. We found that RTK inhibition with a panel of RTK inhibitors decreased cell motility and cell invasion and promoted dramatic actin cytoskeleton reorganisation through activation of the RhoA/Rho-associated protein kinase 1 (ROCK) axis. RTK inhibition also significantly decreased Rnd3 expression levels. Consistently, shRNA-mediated Rnd3 silencing revealed that Rnd3 depletion promoted substantial changes in the actin cytoskeleton and reduced cell motility and invasion capacity, recapitulating the effects observed upon RTK inhibition. Our results indicate that Rnd3 is a crucial mediator of RTK oncogenic signalling involved in actin cytoskeletal reorganisation, which contributes to determining the invasive phenotype of GBM cells.

## 1. Introduction

Malignant gliomas constitute the most frequent primary tumours within the central nervous system (CNS). In their most aggressive form, grade IV glioma or glioblastoma (GBM), these tumours rank among the most devastating human cancers, with an average survival of only 9–15 months [1]. Their dismal prognosis is the result of several biological features, such as enhanced resistance to radio- and chemotherapy and an unparalleled invasive nature, resulting in extensive infiltration into the adjacent brain parenchyma, which ultimately hampers complete surgical resection [2]. These defining biological traits critically challenge the success of therapeutic interventions, highlighting the crucial need for a better understanding of the molecular mechanisms underlying the invasive nature of GBM cells. Recent genomic and transcriptomic analyses have identified the receptor tyrosine kinase (RTK)/Ras/PI3K signalling node as the pathway harbouring the highest number of molecular lesions in GBM [3,4]. Among these, one of the most frequent is the amplification of the epidermal growth factor receptor (EGFR), found in 40–50% of GBMs [5]. Furthermore, almost half of GBMs presenting amplified EGFR co-express the EGFRvIII mutant variant, a truncated receptor lacking a portion of the ligand-binding extracellular domain that is constitutively active [6]. Moreover, other RTKs have been found to be hyperactivated in GBMs, including the platelet-derived growth factor receptor (PDGFR) [7], the vascular endothelial growth factor receptor (VEGFR) [8] and the hepatocyte growth factor receptor (HGFR), also termed c-Met [9]. Collectively, enforced RTK activation in GBM critically contributes to its malignant traits, including increased invasiveness [10], and many receptor tyrosine kinase inhibitors have been tested in the clinical setting either as monotherapy or in combination with other agents, including EGFR, PDGFR and VEGFR inhibitors (reviewed in [10]). However, the precise molecular mechanisms whereby RTK signalling drives GBM cell invasion are still poorly understood.

Increased cell motility, a prerequisite for tumour cell invasion, is dependent on spatiotemporal orchestration of actin cytoskeleton dynamics, a process regulated by Rho GTPases in eukaryotic cells [11]. These proteins function as molecular switches that cycle between an inactive GDP-bound form and an active GTP-bound form. RhoA and Rac1, two prototypic members of this family, control formation of actomyosin fibres (stress fibres) and focal adhesions [12] and lamellipodia and membrane ruffles [13], respectively. A number of reports suggest a relevant role of Rho GTPases in regulation of glioma cell motility and invasion, reviewed in [14]. Specifically, whereas Rac1 exerts a pro-invasive role in GBM [15], activation of RhoA and the consequent increase in actomyosin contractility have been associated with lower motility and invasion [16]. Other members of this family, such as Cdc42 or RhoG, have also been involved in GBM cell invasion [17,18], although the function of many Rho GTPases is still unknown in this context.

Rnd proteins (Rnd1, Rnd2 and Rnd3, also termed RhoE) constitute a separate subfamily within the Rho GTPase family that has attracted recent interest due to their atypical properties and emerging functions [19]. Due to their inability to hydrolyse GTP [20], these proteins are permanently bound to GTP, and, thus, their regulation relies on mechanisms other than the canonical GTP/GDP cycle, such as control of their gene expression, post-translational modifications or subcellular localisation [21]. The best-characterized member of this family, Rnd3, is involved in several cellular processes, including actin cytoskeleton modulation and cell migration. In this context, Rnd3 antagonizes the function of RhoA by activating p190 RhoGAP [22] and also inhibits ROCK directly [23], leading to disassembly of actin stress fibres and focal adhesions and thus modulating cell migration [24]. Rnd3 also influences cell proliferation, with most reports describing an antiproliferative effect [25,26]. In agreement with its cellular functions, a number of studies describing both a pro-tumorigenic and a tumour suppressor role have been reported, largely depending on tumour context, and several reports have described either Rnd3 upregulation or downregulation depending on the type of tumour (reviewed in [27]). In GBMs, Rnd3 is upregulated when compared to low-grade gliomas, although its impact on cell migration and invasion is still unclear owing to contradictory data [28,29].

Here, we have investigated the specific contribution of Rnd3 to GBM cell invasion in the context of RTK and Rho GTPase signalling crosstalk. Using GBM cell lines and patient-derived primary cells, we show that RTK inhibition leads to a decrease in GBM cell motility and invasion, in agreement with actin cytoskeletal reorganization marked by an increase in stress fibre and focal adhesion assembly. RhoA/ROCK inhibition prevents this response, indicating that the effects of RTK inhibition on GBM cell motility and the actin cytoskeleton are mediated through this pathway. Interestingly, RTK signalling is required to sustain Rnd3 expression in GBM cells. In agreement with this and with its antagonistic effects on RhoA/ROCK signalling, Rnd3 overexpression prevents stress fibre formation in cells treated with small-molecule tyrosine kinase inhibitors (TKIs), suggesting that Rnd3 expression contributes to maintaining the invasive phenotype of GBM cells. Accordingly, we show that Rnd3 silencing reduces GBM cell motility and invasion, resulting in cytoskeletal reorganization characterised by increased actomyosin contractility and cellular adhesion that recapitulates the phenotype observed upon RTK inhibition. Our results indicate that Rnd3 is an important downstream mediator of RTK signalling, promoting the invasive phenotype of GBM cells through inhibition of RhoA/ROCK-mediated actin cytoskeleton remodelling.

## 2. Materials and Methods

### 2.1. Cell Culture and Drug Treatment

LN229 and U87MG cells were a gift from Joan Seoane (Vall d’Hebron Institute of Oncology, VHIO, Barcelona, Spain) and were authenticated by short tandem repeat DNA fingerprinting performed by the University of Arizona Genetics Core (University of Arizona, Tucson, AZ, USA). LN229 are characterized by TP53 gene mutation, homozygous p16 and p14ARF deletion and wild-type PTEN, whereas U87MG cells have a wild-type TP53 gene and a mutated PTEN gene background with homozygous p16 and p14ARF deletion [30]. All cell lines were subconfluently grown and passaged and tested for mycoplasma contamination using Mycoalert™ Kit (Lonza, Basel, Switzerland). Cells were grown in DMEM (Sigma-Aldrich, St. Louis, MO, USA) supplemented with 10% foetal calf serum (FCS) (Sigma-Aldrich, St. Louis, MO, USA) in a humidified incubator at 37 °C with 5% CO_2_. For primary GBM cell cultures, samples were obtained from the Biobank at the Institut d’Investigació Sanitària de les Illes Balears (IdISBa) following surgical treatment at the Hospital Universitari Son Espases (Palma, Spain). Written informed consent was obtained from all patients by the biobank and the study was approved by the regional ethical committee (“Comitè d’Ètica de la Investigació-Illes Balears”, CEI, ref. nº IB4132/20PI). Briefly, tumour fragments were washed in phosphate-buffered saline (PBS) and mechanically dissociated into ~1–3 mm^3^ pieces. The pieces were incubated with Accutase Solution (Sigma-Aldrich, St. Louis, MO, USA) for 15 min, and mechanical trituration with a P200 tip was performed every 5 min for further tissue dissociation. The resulting cell suspensions were washed in complete DMEM culture medium and spun 3 min at 300× *g*. After centrifugation, supernatant was discarded and pellets were resuspended in culture medium. Cellular suspensions were seeded in 6-well plates and maintained in a humidified incubator at 37 °C with 5% CO_2_. The patient-derived cell culture used herein (termed P01) was obtained from a single sample from a surgical resection of a recurrent IDH1 wild-type glioblastoma diagnosed case (with 40% ki67).

For drug treatments, erlotinib (Sigma-Aldrich, St. Louis, MO, USA), sunitinib (Sigma-Aldrich, St. Louis, MO, USA), SU-11274 (Sigma-Aldrich, St. Louis, MO, USA) and H-1152 (Calbiochem, Darmstadt, Germany) were added directly to the media at the indicated concentrations, and cells were harvested and analysed at the time points indicated in the figure legends.

### 2.2. Expression Vectors, Transfection and shRNA-Mediated Gene Silencing

Cells were transfected using Lipofectamine2000 (Life Technologies, Carlsbad, CA, USA) according to the manufacturer’s instructions. The following expression plasmids were used: the expression vector encoding Green Fluorescence Protein (GFP) GFP-Rnd3 (Addgene; #23229) and pmaxGFP^®^ (Lonza, Basel, Switzerland). GFP-Rnd3 was a gift from Channing Der (Addgene plasmid #23229; http://n2t.net/addgene:23229 (accessed on 27 January 2021); RRID: Addgene_23229).

For generation of Rnd3-depleted cells, cells were transfected with the expression vectors TRCN0000047713 or TRCN0000047717 from the MISSION^®^ shRNA expression vector library (Sigma-Aldrich, St. Louis, MO, USA). P01 and LN229 cells were reversely transfected in 6-well plates with 1 µg construct using Lipofectamine2000 (Life technologies, Carlsbad, CA, USA) following the manufacturer’s instructions. After 48 h, selection was started by addition of 0.5 µg/mL puromycin (Sigma-Aldrich, St. Louis, MO, USA) to the growth medium, followed by single-cell cloning and expansion. The efficiency of shRNA-mediated silencing was assessed using Western blotting.

### 2.3. Cell Proliferation

Cell growth curve analyses were performed in order to assess differences in cell proliferation. The indicated cell lines were seeded in triplicate into 12-well plates at a density of 2 × 10^4^ cells per well. Cells on each well were counted manually using a Neubauer chamber. Each well was counted thrice at the following time points: 0, 24, 48, 72, 96 and 120 h. Cell media were replaced every 24 h in order to avoid nutrient depletion.

### 2.4. Motility and Invasion Assays

For 2D motility (wound healing assays), 3 × 10^5^ cells were seeded in 6-well plates. After 24 h, the cell monolayer was wounded using a sterile P200 tip, drug treatments were added if needed and at least four representative images of the wound were taken. Images of the same region were collected after 16 (U87MG) or 24 h (LN229 and P01), and the number of cells migrating into the wound was quantified by manually counting the cells present in the wound at the stated times using the Multi-Point tool from Fiji software (ImageJ, v1.8.0). (The results shown here have not been normalized to total cell number since such normalization, tested to rule out the impact on cell proliferation, had negligible effects.)

For 3D invasion assays, 2 × 10^4^ P01 cells were seeded on Matrigel-coated transwells (Corning, Corning, NY, USA) containing DMEM without FCS and placed in 24-well plates containing 10% FCS to create a growth factor gradient. Further, 24 h later, non-invading cells were removed with a cotton swab, and the inserts were transferred into wells with 3.7% (*v*/*v*) paraformaldehyde for fixation and 0.5% (*w*/*v*) crystal violet in 70% ethanol for staining. Images of invading cells from at least three representative fields were taken and quantified by manually counting the number of crystal-violet-stained cells using the Multi-Point tool from Fiji software (ImageJ, v1.8.0, National Institutes of Health, Bethesda, MD, USA).

### 2.5. Immunofluorescence

Cells seeded at low density on coverslips were fixed in 3.7% (*v*/*v*) paraformaldehyde for 20 min and then permeabilized in 0.2% (*v*/*v*) Triton X-100 for 5 min, followed by 1 h incubation in blocking solution (3% bovine serum album in PBS). Cells were then incubated for 24 h at 4 °C with anti-vinculin (1:50 dilution; V4505; Sigma-Aldrich, St. Louis, MO, USA) diluted in blocking solution. Detection was performed with Alexa Fluor 488 donkey anti-mouse (1:1000 dilution, Invitrogen, Carlsbad, CA, USA) diluted in blocking solution and incubated for 1 h at 37 °C. Nuclei were visualized with 4′,6′-diamidino-2-phenylindole (DAPI; 1:2000 dilution; Sigma-Aldrich, St. Louis, MO, USA), and F-actin was visualized with TRITC-phalloidin (1:2000 dilution; Sigma-Aldrich, St. Louis, MO, USA) incubation alongside secondary antibodies. Coverslips were mounted using Fluorescent Mounting Media (DAKO, Glostrup, Denmark). Stained cells were analysed on a Leica TCS SPE confocal microscope (Leica Microsystems, Wetzlar, Germany).

Image analysis was performed using Fiji software (ImageJ, v2.1.0/1.53q, National Institutes of Health, Bethesda, MD, USA). Cell area and circularity analysis were performed by creating a mask for cell area based on F-actin staining and measuring the area and the shape descriptors. Similarly, focal adhesion analysis was performed by creating a mask for focal adhesions based on vinculin staining and applying a minimum size threshold of 0.25 µm^2^ to the Analyse particles command. As for stress fibre analysis, percentage of cells with robust stress fibres was assessed by quantifying the number of cells with robust stress fibres out of the total of cells. Cells were considered to present robust stress fibres when they presented thick, parallel stress fibres across the cell cytoplasm. Complementarily, F-actin intensity profiles were obtained by drawing a straight line either perpendicular to the front-to-rear edge of the cell when inferable or along the shorter axis of the cell and then plotting the intensity profile by using the “Plot Profile” tool from Fiji Software, which displays a two-dimensional graph of the intensities (Mean Gray Value) of pixels along the line (from a to b).

### 2.6. Gel Electrophoresis and Immunoblotting

For immunoblotting, cells were harvested in a lysis buffer containing 50 mM Tris-HCl pH 7.4, 150 mM NaCl, 1 mM EDTA and 1% (*v*/*v*) Triton X-100, protease inhibitors (cOmplete™, Mini Protease Inhibitor Cocktail; Roche Diagnostics, Basel, Switzerland) and phosphatase inhibitors (PhosSTOP™; Sigma-Aldrich; St. Louis, MO, USA).

Protein content was measured using the DC Protein Assay (Bio-Rad, Hercules, CA, USA). Cell lysates were electrophoresed using SDS polyacrylamide gels. After electrophoresis, the proteins were transferred to Immobilon P-strips (Millipore, Billerica, MA, USA) for 60 min at 90 V. The resulting blots were preincubated in TBS (20 mM Tris-HCl pH 7.5, 150 mM NaCl), 0.05% Tween 20 and 5% defatted milk powder for 1 h at room temperature and then incubated overnight at 4 °C with 1% BSA in TBS containing the appropriate antibodies: phospho-Erk(1/2) (Thr202/Tyr204) (#9101), Rnd3 (#3664), Cofilin1 (#5175) and phospho-Cofilin1 (Ser3) (#3313) were from Cell Signaling Technology (Beverly, MA, USA); Erk(1/2) (#MAB15761) was from R&D Systems (Minneapolis, MN, USA) and anti β-tubulin (T0198) was from Sigma-Aldrich (St. Louis, MO, USA). After washing thrice in TBS, 0.05% Tween 20, the blots were incubated with a peroxidase-coupled secondary antibody (Dako, Glostrup, Denmark) in TBS, 0.05% Tween 20, 2% defatted milk powder for 1 h at room temperature. After incubation, the blots were washed thrice in TBS, 0.05% Tween 20 and once in TBS. The peroxidase reaction was visualized using an enhanced chemiluminescence detection system (Millipore, Billerica, MA, USA).

### 2.7. Measurement of Rho GTPase Activity

The amount of active Rho GTPase was analysed based on the capacity of Rho-GTP to bind GST-Rhotekin using the Rho activation assay kit (#BK036, Cytoskeleton, Denver, CO, USA) according to the manufacturer’s instructions. Briefly, 8 × 10^5^ cells were lysed and cleared (10,000× *g*), and a small fraction was separated for total Rho quantification. The remaining bound fraction was incubated in rotation for 1 h at 4 °C with glutathione sepharose-4B beads coupled with GST-Rhotekin for RhoA pulldown. Beads were washed 4 times in wash buffer and spun down. Bound proteins were solubilized by addition of 25 µL of Laemmli loading buffer and separated on 12% SDS-PAGE gels. The amount of Rho-GTP and total Rho were detected by Western blotting with specific antibodies.

### 2.8. Statistical Analysis

All datasets were analysed and plotted using Graphpad Prism (v7.00, Graphpad Software Inc., La Jolla, CA, USA). After testing groups for normal distribution, differences between conditions were analysed using the appropriate statistical test, as stated in the figure legends. Unless stated otherwise, error bars represent SEM.

## 3. Results

### 3.1. Receptor Tyrosine Kinase Inhibition in GBM Cells Reduces Cell Motility and Invasion and Promotes Actin Cytoskeleton Reorganization

In order to characterize the effects of RTK inhibition upon cell motility, we performed wound healing assays in primary glioblastoma cells derived from a patient sample (p01) treated with a panel of small-molecule TKIs: EGFR inhibitor erlotinib [31], PDGFR, VEGFR, Kit and FLT3 multitargeted inhibitor sunitinib [32] and c-Met/HGFR inhibitor SU11274 [33]. Each of these specific compounds was chosen as a well-established inhibitor with the ability to target the aforementioned RTKs, which have all been involved in gliomagenesis [6,7,8,9], and, as readout of their activity in control experiments, they all promoted a clear reduction in phospho-ERK levels in U87MG cells (Appendix A). Meanwhile, untreated cells displayed high motility and migrated very efficiently, showing complete gap closure after 24 h (Figure 1A). RTK inhibition clearly reduced the cell motility rate in these assays (Figure 1A,B). Similarly, cell motility inhibition in response to all three TKIs was observed in U87MG and LN229 cells (Figure 1B and Appendix A). Given that malignant glioma cells display increased motile and invasive capacity, we also investigated whether RTK inhibition with erlotinib, sunitinib or SU11274 could reduce glioma cell invasiveness in a three-dimensional context. Using Matrigel-coated transwells, we observed that, in agreement with our previous results, untreated primary glioma cells (p01) were highly invasive, whereas cell invasion was strongly inhibited in TKI-treated cells (Figure 1C and Figure 2D).

Since fine-tuning of actin cytoskeleton dynamics is required for effective cell motility and invasion, we also performed F-actin and vinculin staining in order to examine the effects of RTK inhibition upon actin cytoskeleton reorganization. F-actin staining indicated that P01 untreated cells were elongated and showed front-to-rear polarity (Figure 2A). In contrast, TKI-treated cells displayed a rigid, firmly attached, well-spread phenotype (Figure 2A), consistent with increased cell area (Figure 2B) and increased actomyosin contractility, as indicated by the rise in the percentage of cells showing robust actin stress fibres (Figure 2C). Additionally, vinculin immunostaining showed that focal adhesions in the control group were reduced both in number (Figure 2D) and area (Figure 2E) and were mostly found on the leading edge of the cell. Meanwhile, in TKI-treated cells, there was an increase in both focal adhesion number and area, and they were scattered throughout the cell surface (Figure 2F). Furthermore, the TKI-induced formation of numerous thick, parallel stress fibres across the cell cytosol is further captured by the F-actin intensity profile (Figure 2F,G).

Similar results were obtained analysing the response of LN229 and U87MG cells (Appendix A). These results indicate that RTK inhibition reduces cell motility and invasion in correlation with dramatic reorganization of the actin cytoskeleton.

### 3.2. Receptor Tyrosine Kinase Inhibition in GBM Cells Regulates Cell Motility and Actin Cytoskeleton Reorganization through Modulation of Rho/ROCK Activity

The effects observed on cell motility and actin cytoskeleton dynamics suggest that RTK inhibition modulates Rho/ROCK signalling to promote formation of actin stress fibres and focal adhesions. In order to test this hypothesis, we performed GST-Rhotekin pull-down assays in P01 cells treated with the same panel of TKIs (Figure 3A). Activation of the RhoA/ROCK pathway was further confirmed in U87MG cells by analysing cofilin activation, a downstream target of ROCK that severs actin filaments. RTK inhibition induced robust cofilin phosphorylation at serine 3, which negatively regulates its activity, thus preventing actin depolymerization (Figure 3B,C).

In order to assess whether the effects of RTK inhibition upon cell motility were conducted through the RhoA/ROCK signalling pathway, we performed wound healing assays with the same panel of TKIs in the absence or in the presence of a small-molecule inhibitor of ROCK (H-1152). As shown previously, TKI treatment decreased cell motility in p01 cells. However, ROCK inhibition restored cell motility in TKI-treated cells to similar rates to that of untreated cells (Figure 3D,E). These results were also reproduced in U87MG cells (Appendix A).

We next analysed F-actin staining in order to evaluate if the effects observed on cell motility correlated with the reorganization of the actin cytoskeleton. In accordance with our previous data, RTK inhibition induced a more rigid, spread phenotype, with robust and thick stress fibres. In contrast, cells co-treated with TKIs and the ROCK inhibitor recovered a thin, elongated phenotype resembling that of untreated cells (Figure 3F), with a smaller cell area (Figure 3G). ROCK inhibition also prevented formation of robust actin stress fibres (Figure 3H) and induced formation of lamellipodia (Figure 2F, blue arrows) and long cell prolongations, consistent with a decrease in the circularity index (Figure 3I). Similar results were obtained in U87MG cells (Appendix A).

Taken together, these results suggest that the effects observed on cell motility and cytoskeleton reorganization upon RTK inhibition are mediated through modulation of the RhoA/ROCK signalling pathway.

### 3.3. Rnd3 Is a Downstream Target of RTK Signalling That Participates in Actin Cytoskeleton Reorganization

Rnd3 is a member of the Rnd subfamily of atypical Rho GTPases that is overexpressed in several types of tumours, including glioblastoma [27], and has been shown to negatively regulate Rho/ROCK activity [22,23,34]. In order to assess whether RTK inhibition affected Rnd3 expression, we treated primary GBM cells, U87MG and LN229 with the same panel of TKIs and measured the Rnd3 expression levels by Western blotting. Interestingly, the Rnd3 expression levels decreased upon RTK inhibition in all the cell lines (Figure 4A,B), suggesting that enforced RTK signalling in GBM cells is key to sustain Rnd3 expression.

Since Rnd3 has been widely reported to inhibit actin stress fibre formation [19,24], we evaluated the effects of Rnd3 overexpression on actin cytoskeleton organization in GBM cells. To this end, we performed F-actin staining in GFP-Rnd3 transfected cells, either untreated or treated with RTK inhibitors. Control GFP-transfection did not discernibly affect cell morphology or stress fibre formation in p01 cells (Figure 4C). In contrast, GFP-Rnd3 transfected cells showed a rounded and elongated phenotype, consistent with smaller cell area (Figure 4C,D). In TKI-treated cells, Rnd3 overexpression prevented formation of robust stress fibres induced by RTK inhibition, with transfected cells resembling phenotypically those of the control group (Figure 4C). These results were also reproduced in U87MG and LN229 cells (Figure 4D and Appendix A). Taken together, these observations identify Rnd3 as a relevant downstream target of RTK signalling in GBM cells endowed with the ability to modulate Rho/ROCK-dependent actin cytoskeleton organization.

### 3.4. Rnd3 Silencing Recapitulates the Effects of RTK Inhibition on Cell Motility and Actin Cytoskeleton Organization

We finally investigated the specific contribution of Rnd3 in the maintenance of the invasive phenotype of GBM cells. For this purpose, cells were transfected with Rnd3-specific (shRnd3) constructs in order to generate Rnd3-silenced primary cells (p01) and LN229 cells (Figure 5A,B). Since Rnd3 has been reported to regulate different cellular functions, such as cell proliferation, in order to verify whether these effects could interfere with migration and invasion assays, we assessed whether cell proliferation was affected in Rnd3 depleted cells. However, cell proliferation assays showed that, although Rnd3 silencing led to an increase in cellular growth, this was only significant after 72 h in P01 and after 96 h in LN229 cells (Appendix A). We, therefore, analysed the effects of Rnd3 silencing upon cell motility, invasion and actin cytoskeleton organization, which were conducted at 24 h time points. Similar to the effects observed in response to TKIs, Rnd3 silencing clearly reduced cell motility in p01 and LN229 cells (Figure 5C,E and Appendix A) and dramatically reduced cell invasion in p01 cells (Figure 5D,F).

As expected, actin staining revealed that Rnd3-depleted cells were significantly larger (Figure 5G,H and Appendix A) and presented thick and robust actin stress fibres (Figure 5G,I and Appendix A). Accordingly, vinculin staining showed that focal adhesions were both increased in number per cell (Figure 5J) and area (Figure 5K) and were widely distributed across the cell surface (Figure 5L). Intensity profile graphs of F-actin staining further illustrate the stress fibre pattern formed upon Rnd3 depletion, characterised by an increase in the number of thick, parallel stress fibres throughout the cell body (Figure 5L,M and Appendix A).

In order to assess whether the effects on the actin cytoskeleton elicited by Rnd3 silencing were also mediated through increased RhoA/ROCK signalling, we performed F-actin staining in Rnd3-depleted cells treated with ROCK inhibitor H-1152 (Figure 6A). Although ROCK inhibition in Rnd3-depleted cells did not cause a remarkable effect on the cell area (Figure 6B), it significantly reduced the number of cells with robust stress fibres (Figure 6C). ROCK inhibition also led to formation of long cell prolongations and lamellipodia (Figure 6A, blue arrows), as reflected in a decrease in the circularity index (Figure 5D). Altogether, these results indicate that Rnd3 contributes to actin cytoskeleton remodelling and maintenance of a highly motile phenotype.

## 4. Discussion

Increased GBM cell invasiveness is a crucial biological hallmark severely jeopardizing the clinical outcome of these malignant tumours. For this reason, we have investigated its underlying molecular mechanisms, focusing specifically on the functional crosstalk between RTKs activated in GBM and Rho GTPases, considering their central role in the control of actin cytoskeleton regulation and cell migration, also in GBM [14]. In this regard, the accumulated evidence so far indicates that Rac1 activation promotes GBM cell invasion [15], whilst activation of Rho leads to a reduction in cell motility and invasion [16]. However, the precise mechanisms whereby oncogenic signalling emanating from RTKs promotes the invasive phenotype of GBM cells are largely unknown, although some mediators have been proposed to play an important role, such as Rac guanine-nucleotide exchange factor (GEF) Dock180 that has been shown to be activated downstream of EGFR, PDGFR and c-Met [35,36,37].

The collective influence of RTK signalling on cell motility and invasion was validated with our observations, showing that, both in primary patient-derived cells and in established GBM cell lines, inhibition of RTKs, including EGFR, PDGFR, VEGFR and c-Met, promoted a clear reduction in both cell motility and invasion. Importantly, this reduction in cell migration was also accompanied by a dramatic reorganization of the actin cytoskeleton in TKI-treated cells, characterized by assembly of robust actin stress fibres, an increase in the size and number of focal adhesions and modification of cellular morphology from spindle-shaped to a larger polygonal form. These cytoskeletal changes clearly bear the hallmarks of Rho/ROCK activation [12], and, accordingly, our data confirmed that RTK inhibition in GBM cells leads to Rho/ROCK pathway activation. Crucially, our data also indicate that activation of the Rho/ROCK pathway is strictly required to promote these cellular effects downstream of RTK inhibition. These results extend and confirm the notion that activation of Rho signalling inhibits GBM cell motility and invasion through reorganization of the actin cytoskeleton, as also reported in GBM cells treated with resveratrol [38].

As mentioned earlier, a number of reports have focused on molecular mechanisms leading to Rac activation in response to RTK activation in GBM. However, there is less information on the mechanisms regulating Rho/ROCK activation in this same context other than presumably operating compensatory feedback between Rac and Rho [39]. Noteworthy, in another highly invasive cancer such as melanoma, B-RAF inhibition similarly reduces cell motility and invasion through Rho/ROCK-dependent actin cytoskeletal reorganization. Interestingly, in this setting, atypical Rho GTPase family member Rnd3 was identified as a critical regulator of crosstalk between B-RAF and Rho/ROCK signalling pathways since B-RAF-sustained Rnd3 expression is essential for acquisition of the invasive phenotype through inhibition of Rho/ROCK [40]. We thus reasoned that RTK signalling in GBM might also influence Rho/ROCK function through modulation of Rnd3 expression levels since Rnd3 has been shown to be moderately overexpressed in GBM and is a widely reported Rho/ROCK functional antagonist [22,23,34]. Inhibition of RTKs revealed that Rnd3 expression is indeed under the control of RTK signalling, and, accordingly with its established effects, moderate Rnd3 overexpression is able to prevent TKI-induced cytoskeletal reorganization, suggesting that Rnd3 downregulation upon RTK inhibition is the key event mediating these cytoskeletal effects. Conversely, Rnd3 silencing essentially recapitulated all the observed effects promoted by RTK inhibition on actin cytoskeletal dynamics and cell motility, thus indicating that Rnd3 is a crucial mediator of the invasive phenotype in GBM cells.

These results are in agreement with others describing Rnd3 as a key gene establishing multiple hallmarks of cancer in GBM, in which Rnd3 was found to be upregulated in high-grade gliomas through analysis of multiple network models and is a key regulator of migration and invasion [28]. Importantly, Rnd3 expression was also found to be predictive of clinical outcome. Surprisingly, however, in this setting, Rnd3 upregulation was also associated with increased cell proliferation, in contrast with our observation that Rnd3 silencing promotes a faster rate of cell proliferation, in agreement with previous reports indicating that Rnd3 exerts an antiproliferative effect in G_1_ progression, also in GBM cells [25,26]. Strikingly, this controversy is fuelled by other reports investigating Rnd3 in GBM and describing the opposite functions and expression levels. In U251 cells, Rnd3 was associated with inhibition of migration, invasion and also proliferation, and its expression levels were found to be downregulated in GBM specimens, as shown by immunohistochemistry [29]. Apparently, Rnd3 exerts these effects through suppression of the Notch signalling complex [41]. Such conflicting data regarding the cellular functions of Rnd3 in GBM, most likely related to the methodology and experimental models used in different studies, clearly highlight the need for further investigation on Rnd3 gene function and regulation in GBM. For instance, little is known about the precise regulation of Rnd3 expression in GBM, which has a clear impact on its function as Rnd proteins do not hydrolyse GTP [21]. It is likely that specific signalling pathways engaged by RTKs in GBM cells are responsible for Rnd3 sustained expression levels by promoting activation of certain transcription factors. In this regard, experiments addressed at elucidating RTK-dependent signalling pathways that could drive Rnd3 expression in malignant GBM are currently underway in our laboratory.

In our model, highly invasive GBM cells are characterized by a spindle-shaped cellular morphology and a “fluid” cytoskeletal configuration dependent on a high Rac/low Rho activity balance that is maintained, at least partially, by sustaining Rnd3 expression. Disruption of RTK signalling, similar to Rnd3 silencing, leads to Rnd3 downregulation and produces an imbalance caused by the robust activation of the Rho/ROCK pathway, which results in a larger and firmly attached cellular morphology and a “rigid” cytoskeletal configuration, thus reducing cell migration (Figure 7). Taken together, our data identify Rnd3 as a crucial mediator of RTK-driven GBM cell invasiveness.

## Figures and Tables

**Figure 1 cells-11-03716-f001:**
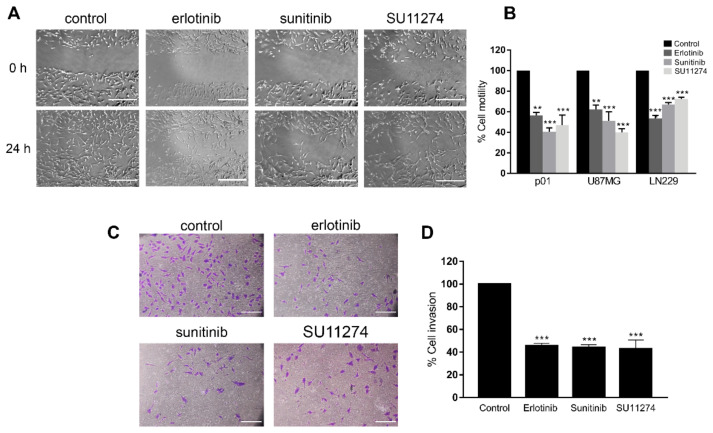
Receptor tyrosine kinase inhibition in GBM cells reduces cell motility and invasion. (**A**) Representative phase-contrast micrographs of P01 cells treated either with vehicle (control) or 10 µM erlotinib, 2.5 µM sunitinib or 2.5 µM SU11274 as indicated at 0 h (upper panel) and 24 h after (lower panel) performing wound healing assays as described in Materials and Methods. Scale bar, 500 µm. (**B**) Representation of the mean ± SEM rate of motility of the indicated GBM cells from three independent experiments, expressed as the percentage of cell motility relative to control cells. Statistical significance was assessed by one-way analysis of variance (ANOVA) with Dunnett’s multiple comparisons test. (**C**) Representative images of crystal-violet-stained P01 cells after performing Matrigel-coated transwell invasion assay. Cells were seeded onto Matrigel-coated transwells, treated either with vehicle (control) or 10 µM erlotinib, 2.5 µM sunitinib or 2.5 µM SU11274 as indicated and incubated for 24 h. Scale bar, 250 µm. (**D**) Representation of the mean ± SEM rate of invasion from three independent experiments performed in duplicate, expressed as the percentage of cell invasion relative to control cells. Statistical significance was assessed by one-way analysis of variance (ANOVA) with Dunnett’s multiple comparisons test. Statistical significance is indicated as follows: ** *p* < 0.01, *** *p* < 0.001.

**Figure 2 cells-11-03716-f002:**
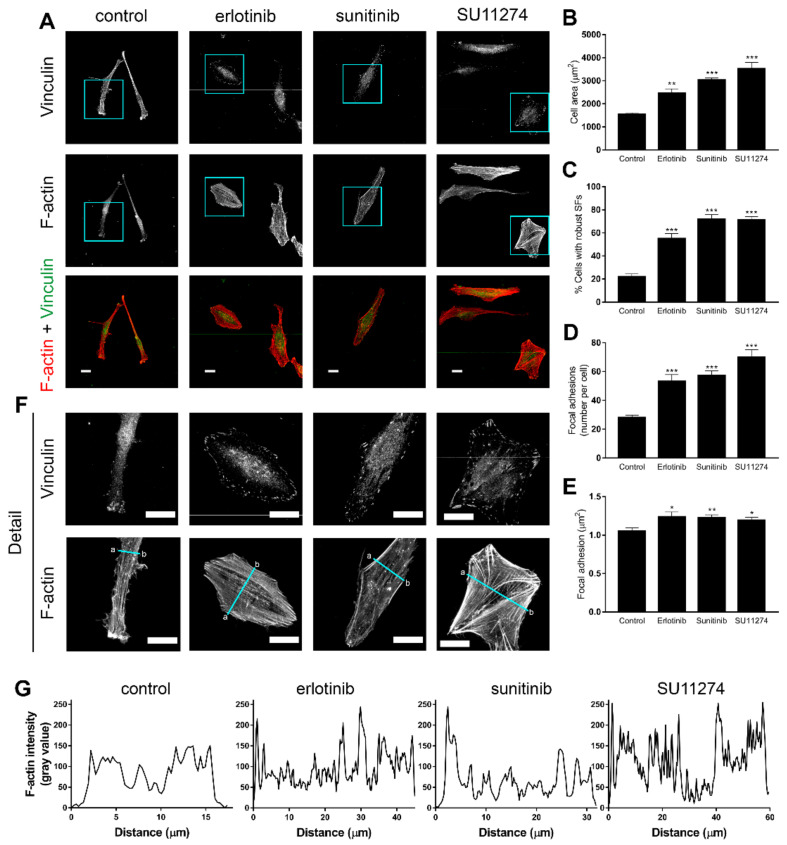
Receptor tyrosine kinase inhibition in GBM cells promotes actin cytoskeleton reorganization and focal adhesion assembly. (**A**) P01 cells were grown on coverslips and treated for 24 h with either vehicle (control) or 10 µM erlotinib, 2.5 µM sunitinib or 2.5 µM SU11274 as indicated, fixed and stained with anti-vinculin antibody (vinculin, green) and TRITC-labelled phalloidin (F-actin, red). Scale bar, 25 µm. (**B**) Quantification of cell area (µm^2^) and (**C**) percentage of cells with robust stress fibres (SFs) of P01 cells from (**A**). Values represent the mean ± SEM from five independent experiments. Total number of cells analysed: *n* = 252 control, *n* = 252 erlotinib, *n* = 262 sunitinib, *n* = 273 SU11274. Statistical significance was assessed by one-way analysis of variance (ANOVA) with Dunnett’s multiple comparisons test. (**D**) Quantification of focal adhesion number per cell and (**E**) average focal adhesion area per cell of P01 cells from (**A**). Values represent the mean ± SEM from *n* = 41 control, *n* = 29 erlotinib, *n* = 54 sunitinib and *n* = 44 SU11274 cells. Statistical significance was assessed using a Kruskal–Wallis nonparametric test with Dunnett’s multiple comparisons test. (**F**) Magnified images from the boxed regions in (**A**); scale bar, 25 µm. (**G**) Intensity profiles of F-actin from the blue lines drawn in (**F**) from a to b. Statistical significance is indicated as follows: * *p* < 0.05, ** *p* < 0.01, *** *p* < 0.001.

**Figure 3 cells-11-03716-f003:**
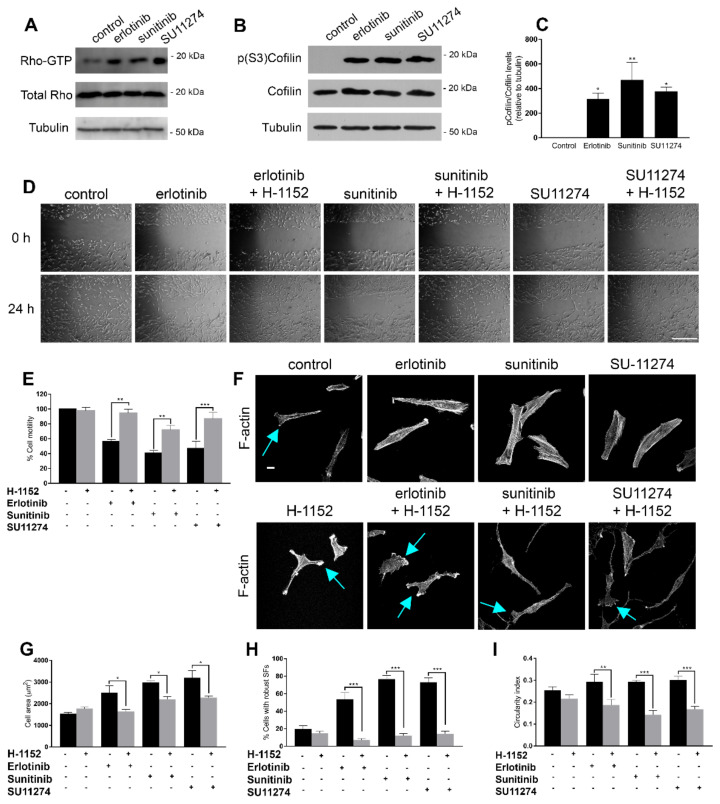
Receptor tyrosine kinase inhibition in GBM cells regulates cell motility and actin cytoskeleton reorganization through Rho GTPase modulation. (**A**) P01 cells were treated with 10 µM erlotinib, 2.5 µM sunitinib or 2.5 µM SU11274 for 24 h, harvested and RhoA activation was analysed by GST-Rhotekin pulldown, followed by Western blotting with anti-RhoA (upper panel). An aliquot of each lysate was also loaded to analyse total RhoA levels (bottom panel). (**B**) U87MG cells were treated with 10 µM erlotinib, 2.5 µM sunitinib or 2.5 µM SU11274 for 16 h, harvested and phospho-cofilin (Ser3) and total cofilin levels were analysed by Western blotting. (**C**) Densitometric quantification of phospho-cofilin (Ser3) levels from (**B**). Values represent the mean ± SEM from three independent experiments. Statistical significance was assessed by one-way analysis of variance (ANOVA) with Dunnett’s multiple comparisons test. (**D**) Representative phase-contrast micrographs of P01 cells treated with vehicle (control) or with 10 µM erlotinib, 2.5 µM sunitinib or 2.5 µM SU11274 alone or in the presence of 0.5 µM H-1152 at 0 h (upper panel) or 24 h after (lower panel) performing wound healing assays as described in Materials and Methods. (**E**) Representation of the mean ± SEM rate of motility of P01 cells from three independent experiments, expressed as the percentage of cell motility relative to control cells. Statistical significance was assessed by one-way analysis of variance (ANOVA) with Dunnett’s multiple comparisons test. (**F**) P01 cells were grown on coverslips and treated for 24 h with vehicle (control) or 10 µM erlotinib, 2.5 µM sunitinib or 2.5 µM SU11274 alone (upper panel) or in the presence of 0.5 µM H-1152 (lower panel), fixed and stained with TRITC-labelled phalloidin (F-actin). Blue arrows, lamellipodia; scale bar, 25 µm. (**G**) Quantification of cell area (µm^2^), (**H**) percentage of cells with robust stress fibres (SFs) and (**I**) circularity index of p01 cells from (**F**). Values represent the mean ± SEM from three independent experiments. Total number of cells analysed: *n* = 168 control, *n* = 196 H-1152, *n* = 145 erlotinib, *n* = 177 erlotinib+H-1152, *n* = 158 sunitinib, *n* = 156 sunitinib+H-1152, *n* = 164 SU11274, *n* = 153 SU11274+H-1152. Statistical significance was assessed by one-way analysis of variance (ANOVA) with Dunnett’s multiple comparisons test. Statistical significance is indicated as follows: * *p* < 0.05, ** *p* < 0.01, *** *p* < 0.001.

**Figure 4 cells-11-03716-f004:**
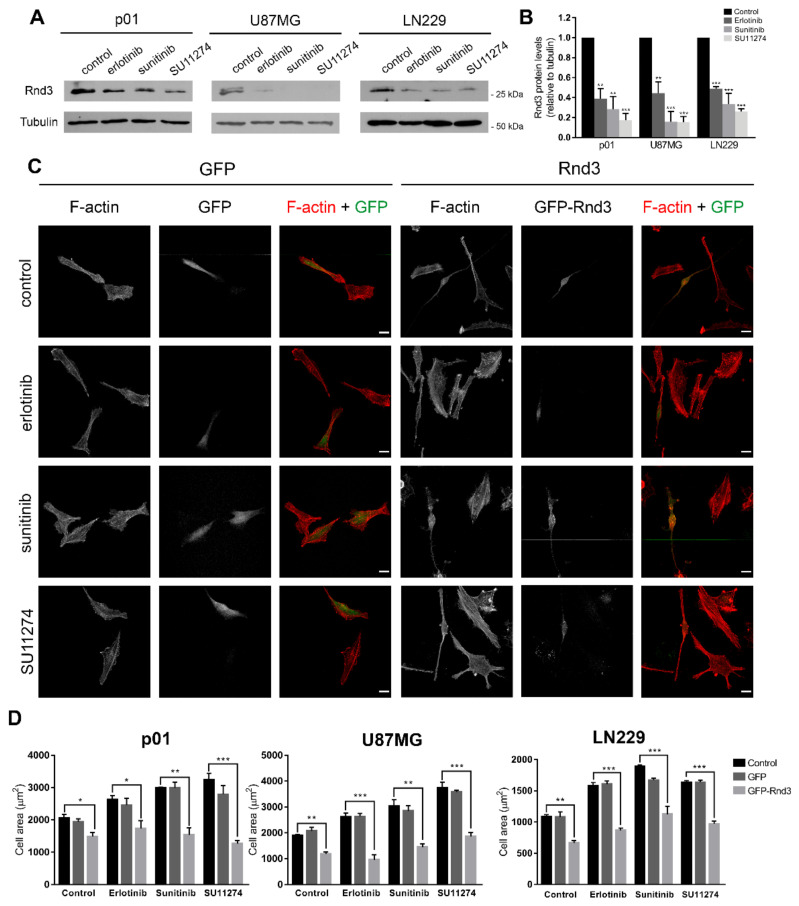
Rnd3 is a downstream target of RTK signalling that participates in actin organization. (**A**) p01, U87MG and LN229 cells were treated with 10 µM erlotinib, 2.5 µM sunitinib or 2.5 µM SU11274 for 24 h (p01, LN229) or 16 h (U87MG), harvested and Rnd3 expression levels were analysed by Western blotting. (**B**) Densitometric quantification of Rnd3 levels from (**A**). Values represent the mean ± SEM from three independent experiments. Statistical significance was assessed by one-way analysis of variance (ANOVA) with Dunnett’s multiple comparisons test. (**C**) p01 cells were transfected with Rnd3-GFP or GFP encoding plasmids, treated for 24 h with 10 µM erlotinib, 2.5 µM sunitinib or 2.5 µM SU11274, fixed and stained with TRITC-labelled phalloidin (F-actin). Scale bar, 25 µm. (**D**) Quantification of cell area (µm^2^) of p01 cells from (**C**), U87MG cells from Appendix A and LN229 cells from Appendix A. Values represent the mean ± SEM from three independent experiments. Total number of cells analysed: P01: control (*n* = 61 control, *n* = 40 GFP, *n* = 31 GFP-Rnd3), erlotinib (*n* = 76 control, *n* = 28 GFP, *n* = 35 GFP-Rnd3), sunitinib (*n* = 40 control, *n* = 30 GFP, *n* = 18 GFP-Rnd3), SU11274 (*n* = 64 control, *n* = 29 GFP, *n* = 26 GFP-Rnd3); U87MG: control (*n* = 136 control, *n* = 37 GFP, *n* = 35 GFP-Rnd3), erlotinib (*n* = 97 control, *n* = 37 GFP, *n* = 16 GFP-Rnd3), sunitinib (*n* = 91 control, *n* = 35 GFP, *n* = 28 GFP-Rnd3), SU11274 (*n* = 71 control, *n* = 33 GFP, *n* = 15 GFP-Rnd3); LN229: control (*n* = 131 control, *n* = 47 GFP, *n* = 34 GFP-Rnd3), erlotinib (*n* = 87 control, *n* = 54 GFP, *n* = 28 GFP-Rnd3), sunitinib (*n* = 64 control, *n* = 35 GFP, *n* = 24 GFP-Rnd3), SU11274 (*n* = 97 control, *n* = 41 GFP, *n* = 24 GFP-Rnd3). Statistical significance was assessed by one-way analysis of variance (ANOVA) with Dunnett’s multiple comparisons test. Statistical significance is indicated as follows: * *p* < 0.05, ** *p* < 0.01, *** *p* < 0.001.

**Figure 5 cells-11-03716-f005:**
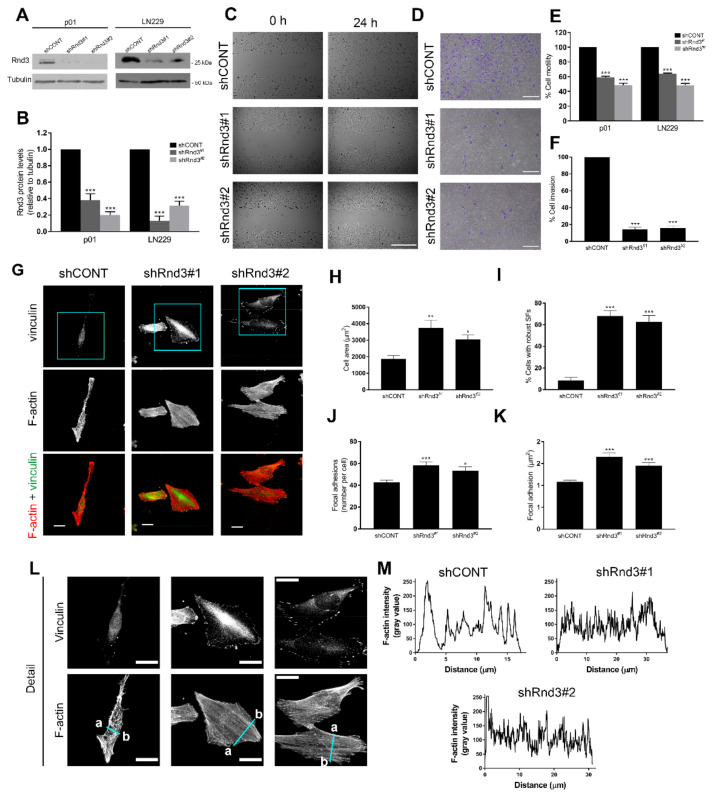
Rnd3 is a crucial mediator of cell motility and invasion and modulates cytoskeleton dynamics in GBM cells. (**A**) p01 and LN229 cells were transfected with a control shRNA (shCONT) or a shRNA targeting human RND3 (shRnd3), after which stable transfectants were generated, as indicated in Materials and Methods. Control and Rnd3-depleted cells were harvested and Rnd3 expression levels were analysed by Western blotting. (**B**) Densitometric quantification of Rnd3 levels from (**A**). Values represent the mean ± SEM from three independent experiments. (**C**) Representative phase-contrast micrographs of control (shCONT) and Rnd3-depleted (shRnd3) p01 cells at 0 h (left panel) and 24 h after (right panel), performing wound healing assays as described in Materials and Methods. Scale bar, 500 µm. (**D**) Representative images of crystal-violet-stained P01 cells after performing Matrigel-coated transwell invasion assay. P01 control (shCONT) and Rnd3-depleted (shRnd3) cells were seeded onto Matrigel-coated transwells and incubated for 24 h. Scale bar, 250 µm. (**E**) Representation of the mean ± SEM rate of motility of the indicated GBM cells from three independent experiments, expressed as the percentage of cell motility relative to control cells. Statistical significance was assessed by one-way analysis of variance (ANOVA) with Dunnett’s multiple comparisons test. (**F**) Representation of the mean ± SEM rate of invasion of P01 cells from three independent experiments performed in duplicate, expressed as the percentage of cell invasion relative to control cells. Statistical significance was assessed by one-way analysis of variance (ANOVA) with Dunnett’s multiple comparisons test. (**G**) P01 control (shCONT) and Rnd3-depleted (shRnd3) cells were grown on coverslips, fixed and stained with anti-vinculin antibody (vinculin, green) and TRITC-labelled phalloidin (F-actin, red). Scale bar, 25 µm. (**H**) Quantification of cell area (µm^2^) and (**I**) percentage of cells with robust stress fibres (SFs) of P01 cells from (**G**). Values represent the mean ± SEM from four independent experiments. Total number of cells analysed: *n* = 130 shCONT, *n* = 125 shRnd3#1, *n* = 118 shRnd3#2. Statistical significance was assessed by one-way analysis of variance (ANOVA) with Dunnett’s multiple comparisons test. (**J**) Quantification of focal adhesion number per cell and (**K**) average focal adhesion area per cell of P01 cells from (**G**). Values represent the mean ± SEM from *n* = 39 shCONT, *n* = 34 shRnd3#1, *n* = 24 shRnd3#2. Statistical significance was assessed using a Kruskal–Wallis nonparametric test with Dunnett’s multiple comparisons test. (**L**) Magnified images from the boxed regions in (**G**); scale bar, 25 µm. (**M**) Intensity profiles of F-actin from the blue lines drawn in (**L**) from a to b. Statistical significance is indicated as follows: * *p* < 0.05, ** *p* < 0.01, *** *p* < 0.001.

**Figure 6 cells-11-03716-f006:**
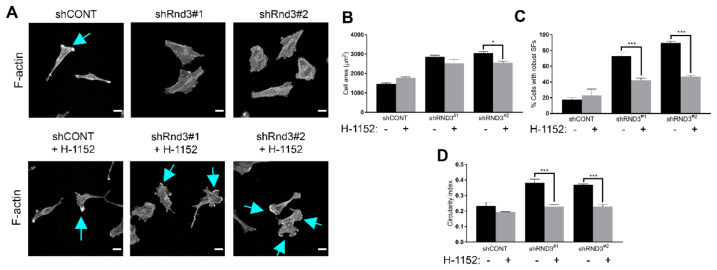
ROCK inhibition induces morphological changes and cytoskeleton remodelling in Rnd3-depleted cells. (**A**) P01 control (shCONT) and Rnd3-depleted (shRnd3) cells were grown on coverslips and treated for 24 h with 0.5 µM H-1152, fixed and stained with TRITC-labelled phalloidin (F-actin). Blue arrows, lamellipodia; scale bar, 25 µm. (**B**) Quantification of cell area (µm^2^), (**C**) percentage of cells with robust stress fibres (SFs) and (**D**) circularity index of p01 cells from (**A**). Values represent the mean ± SEM from three independent experiments. Total number of cells analysed: *n* = 150 shCONT, *n* = 134 shCONT + H-1152, *n* = 150 shRnd3#1, *n* = 167 shRnd3#1 + H-1152, *n* = 162 shRnd3#2, *n* = 149 shRnd3#2 + H-1152. Statistical significance was assessed by one-way analysis of variance (ANOVA) with Dunnett’s multiple comparisons test. Statistical significance is indicated as follows: * *p* < 0.05, *** *p* < 0.001.

**Figure 7 cells-11-03716-f007:**
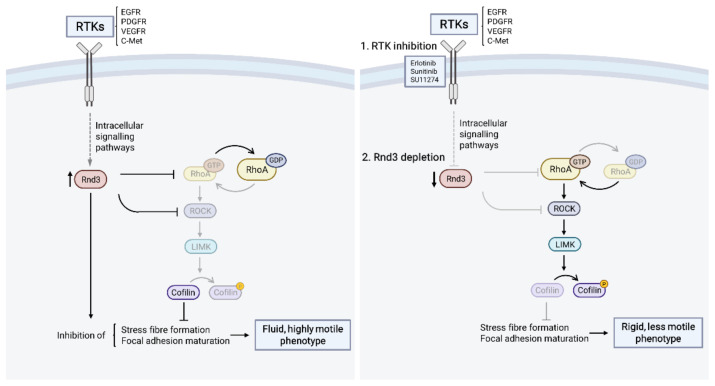
Working model depicting RTK and Rho GTPase signalling crosstalk and its impact on GBM cell invasion. (Left) RTKs are activated through diverse molecular mechanisms, sustaining high Rnd3 expression levels in glioma cells. Rnd3 antagonizes the RhoA/ROCK signalling pathway, leading to low basal RhoA activity and dephosphorylation of ROCK downstream effectors, such as cofilin. This Rho GTPase balance modulates actin cytoskeleton dynamics favouring inhibition of stress fibre formation and focal adhesion maturation, thus leading to a more fluid and highly motile phenotype. (Right) Inhibition of different RTKs with a panel of small-molecule tyrosine kinase inhibitors (1) reduces Rnd3 expression, which, in turn, allows for increased RhoA/ROCK signalling. This switch in Rho GTPase activity orchestrates actin cytoskeleton remodelling, characterised by formation of robust and thick stress fibres, allowed by increased cofilin phosphorylation and focal adhesion maturation, resulting in a more rigid and less motile phenotype. This phenotype is also recapitulated upon Rnd3 shRNA-mediated depletion (2).

## Data Availability

Not applicable.

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
