# Peer review of "Rnd3 Is a Crucial Mediator of the Invasive Phenotype of Glioblastoma Cells Downstream of Receptor Tyrosine Kinase Signalling"

_cells, 2022, doi:10.3390/cells11233716_

Round 1

Reviewer 1 Report (Previous Reviewer 1)

Even if the revised manuscript was not easy to review due to the mixture of old and new figures in the file, the authors have adequately addressed the points initially raised by myself.

Here are the minor points that remain to be fixed:

- Line 244. Since primary GBM cells were derived from only one patient sample (P01), the sentence “primary glioblastoma cells derived from patient samples (p01)” must be corrected and turn to singular.

- Figure S1B: Images from U87MG wound healing assays are too dark and must be brightened.

Author Response

We thank the reviewer for noting these issues, thay have now been modified and corrected in full.

Reviewer 2 Report (Previous Reviewer 3)

The authors responded to all my requests/questions appropriately. 

Author Response

We are pleased that the revised version has satisfied the reviewer.

Reviewer 3 Report (Previous Reviewer 4)

The revised manuscript entitled “Rnd3 is a crucial mediator of the invasive phenotype of glioblastoma cells downstream of receptor tyrosine kinase signaling” by Almarán et al. deals with my criticisms thoroughly and all my original specific points and suggestions were properly addressed. The authors reorganized their figures, and further discussions and references were added as requested. In my opinion the revised manuscript improved significantly.

Author Response

We thank the reviewer for his positive comments.

This manuscript is a resubmission of an earlier submission. The following is a list of the peer review reports and author responses from that submission.

Round 1

Reviewer 1 Report

Almarán et al. characterized the in vitro effects of receptor tyrosine kinase (RTK) inhibition on the invasive capacity of both, primary glioblastoma cells derived from patient samples and immortalized glioblastoma cell lines. They demonstrated that RTK inhibitors lead to a decrease in cell motility and invasion accompanied by a reorganization of the actin cytoskeleton with a robust actin stress fiber assembly and an increase in the size and number of focal adhesions. These effects are dependent on the activation of Rho/ROCK pathway. The authors further identified Rnd3 as a mediator of these RTK inhibitors-mediated phenotypes in GBM cells.

Majority of the data presented in the manuscript are of good technical quality, even if the quality of some images may be increased. Moreover as described below, some issues remain to be addressed.   

  1. In the introduction section, statement about the use of RTK inhibitors in clinics for GBM is missing.
  2. The authors used primary glioblastoma cells derived from patient samples (P01). It is not clear if all P01 cells used in the paper were derived from the same patient or from different patients. What is known about these samples in terms of mutations and RTK signaling activation? Same question for U87MG and LN229.
  3. Proper controls about the efficacy of drugs on their target are missing. The monitoring by western-blot analyses of their efficacy on downstream pathways such as the MAPK pathway for RTK inhibitors and on RhoA/ROCK pathway for H-1152 must be provided for each cell line.
  4. Figure S1. Wound-healing assays must be performed using confluent cells. The conditions for U87MG as visible on figure S1A are not satisfying. Please revise this experiment.
  5. Figure 2. As the paper focuses on the invasiveness of GBM cells, the impact of H-1152 treatment on RTKi-treated cells should be evaluated also using invasion assays.
  6. The authors discuss about the RhoA/Rac balance. What is the status of Rac1 activation upon drug treatment in GBM cells?
  7. Figure 3. As a member of the GTPase family with a CAAX box in Cter, GFP fusions are allowed only in Nter. As described in the Mat & Met section, the fused protein is likely GFP-Rnd3, instead of Rnd3-GFP. Please correct on Figure 3 and in text.
  8. Figure 3. As quality control, please show the expression of GFP-Rnd3 by western blot analysis upon transfection in GBM cells. What is the impact of Rnd3 overexpression on cell motility and/or cell invasion upon treatment with RTKi of GBM cells, i.e. does Rnd3 overexpression rescue RTKi impact on cell migration and/or invasion?
  9. Information about the quantification of migrating cells using wound-healing assay is required.
  10. Information about the qualification and the quantification of “robust stress fibers” is missing.
  11. In the discussion section, please add hypotheses about the way RTKi may regulate Rnd3 expression.
  12. Legends for the supplemental files are missing.

Reviewer 2 Report

Reviewer 1

This original article study is largely confirmatory of a previously published study by Oncotarget. 2016 Dec 13;7(50):82411-82423.; Cell Prolif. 2019 Sep;52(5):e12665.; Cancer Med. 2015 Sep;4(9):1404-16.   Baohui Liu et al., 2016 studies underscore clearly the RND3 promotes Snail 1 protein degradation and inhibits glioblastoma cell migration and invasion, and defines a new function of RND3 in GBM invasive phenotype, and therefore lacks significant novelty of this original study.

Reviewer 3 Report

In this study, the authors described about essential role of receptor tyrosine kinases (RTKs)-Rnd3-Rho/Rock signaling in positive regulation of glioblastoma (GBM) cell invasive phenotype. The authors discovered this molecular axis in GBM cells by using in vitro system.

# Comments:

1)     Because Rnd3 has been already reported to regulate various cellular functions such as cell viability or proliferation, the authors should demonstrate the effect of Rnd3 knockdown in these GBM cellular functions as well as cell motility/invasion.

2)     In immunofluorescence studies, higher magnification photos are also necessary to monitor more accurate intracellular expression pattern of f-actin and vinculin.

3)     The results of cell motility/invasion assay should be corrected by total cell number of each treated cells, because the effects of each treatment against cell proliferation could not be ignored.

4)     The authors should demonstrate activation status of representative RTKs and their downstream signaling under treatment by RTK inhibitors used in this study.

5)     The authors should use multiple Rock inhibitors for the assays.

6)     Because supplementary figure S2G-S2I are considered essential and important for this study, these data should be transferred as the main figures, In addition, the authors should also monitor Rho-GTPase activity under Rnd3 knockdown in GBM cells.

7)     The authors should demonstrate whether activation of RTKs by such as their ligands induces upregulation of Rnd3 expression and cell motility/invasion in GBM cells used in this study.

Reviewer 4 Report

The manuscript entitled “Rnd3 is a crucial mediator of the invasive phenotype of glioblastoma cells downstream of receptor tyrosine kinase signaling” by Almarán et al. is an interesting and a well-written report.

The manuscript has a well-balanced sectional structure with an appropriately detailed Introduction that gives a good background both for the receptor tyrosine kinase signaling and glioblastoma cells, a detailed Materials and Methods section, and a very detailed and convincing Results section. This part is communicated through 4 regular and 4 supplementary figures, all of them include several panels. Although the figures have very high quality, they will not prevail and have less impact than they deserve in their present form because many of their legends are illegible at these figure dimensions. Some of the microscopic images are too small to convey real information. For example, the white arrows indicating lamellipodia in Figure 2 are almost impossible to see. Furthermore, as in Figure 2B, D, E, F, the names of the treatments are too small to read in the expected size in the published manuscript. Perhaps taking apart some of the panels to make a few more independent figures, and making them a bit larger, would help. The Discussion is perhaps a bit shorter than expected but reads well and appropriately places the results within the literature. It has an informative working model (Figure 5) depicting RTK and Rho GTPase signaling crosstalk and its impact on glioblastoma cell invasion. The manuscript has only 37 references; in a vast and rapidly developing field, such a rather limited collection of references seems a bit frugal... I would suggest explaining why these particular drugs (erlotinib, sunitinib, SU-11247 and H-1152) were used. I would add a least one reference for each drug used and explain the choice of applied drug concentrations.

Specific points:

There are several minor problems that the authors should address in their revision.

Although the manuscript is well-written, the text should be thoroughly copy edited for grammar, syntax, and punctuation. This will improve readability and fitness and make the text smoother. I list a few typical mistakes as examples related to misspellings, punctuation problems, etc.:

Lines 6, 7, 9, 99, 107, 110, 500, 503, 504: affiliations are concatenated

Lines 129, 144, 171, 178, 267, 479, 482, 485, 487, 488: wrong spacing of words

Lines 130, 369: RND is fully capitalized only here; is it correct?

Lines 137, 197: Western or western? I suggest the latter one.

Line 498: correctly: submitted version…

The only indication about the number of quantified cells is from the legend to Figure 1. I quote: “Values represent the mean ± SEM from n=41 untreated, n=29 erlotinib, n=54 sunitinib and n=44 SU11274 cells.” Which graph these numbers refer to? Are they for Figure 1 only? How about the data on the other figures (and supplementary figures)? For example, how many cells were quantified to compute the percentage of cells with robust stress fibers? Such data should be placed into the appropriate subsections of the Materials and methods.

Lines 234, 294, 388, 398: how the percentage of cells with robust stress fibers was determined? How to define “robust”? Ideally, this information should be incorporated into the Materials and methods section.

Panel A in Figures 1, 2, and Panel C in Figure 4, should have scale bars. The same is for Figures S1A, S2A and S4A.

Proper figure legends to supplementary figures would be useful (for example, scale bar sizes…).

Graphs are of different sizes. Compare, for example, panel B, I and O in Figure 4. (Panel B is completely unreadable. When enlarged, it becomes pixelated in the pdf.)

Figure 5: My suggestion is: Fluid, highly motile… and Rigid, less motile…

Abbreviations should be explained when used first (PBS, ROCK, Rnd, GFP, LIMK, etc.). TKI is first used in line 90, and not in line 208, where it is written out in full.

A few references are incorrectly cited:

Lines 532–534, 542, 570, 583, 589, 596: journal names are not abbreviated as elsewhere

Line 517: authors’ list is missing from the reference

Line 597: Ref. #38 is missing